# Spatiotemporal Feature Learning Based Hour-Ahead Load Forecasting for Energy Internet

**Liufeng Du [1,2], Linghua Zhang [1,3,*] and Xu Wang [1]**

[1] College of Telecommunications and Information Engineering, Nanjing University of Posts and Telecommunications, Nanjing 210003, China; dulfv@outlook.com (L.D.); wangxhhtc@163.com (X.W.)

[2] School of Mechanical and Electrical Engineering, Henan Institute of Science and Technology, Xinxiang 453003, China

[3] Jiangsu Engineering Research Center of Communication and Network Technology, Nanjing University of Posts and Telecommunications, Nanjing 210003, China

* Correspondence: zhanglh@njupt.edu.cn

**Abstract:** In this paper, we analyze the characteristics of the load forecasting task in the Energy Internet context and the deficiencies of existing methods and then propose a data driven approach for one-hour-ahead load forecasting based on the deep learning paradigm. The proposed scheme involves three aspects. First, we formulate a historical load matrix (HLM) with spatiotemporal correlation combined with the EI scenario and then create a three-dimensional historical load tensor (HLT) that contains the HLMs for multiple consecutive time points before the forecasted hour. Second, we preprocess the HLT leveraging a novel low rank decomposition algorithm and different load gradients, aiming to provide a forecasting model with richer input data. Third, we develop a deep forecasting framework (called the 3D CNN-GRU) featuring a feature learning module followed by a regression module, in which the 3D convolutional neural network (3D CNN) is used to extract the desired feature sequences with time attributes, while the gated recurrent unit (GRU) is responsible for mapping the sequences to the forecast values. By feeding the corresponding load label into the 3D CNN-GRU, our proposed scheme can carry out forecasting tasks for any zone covered by the HLM. The results of self-evaluation and a comparison with several state-of-the-art methods demonstrate the superiority of the proposed scheme.

**Keywords:** load forecasting; Energy Internet; low rank decomposition; 3D CNN; GRU

## 1. Introduction

Accurate and stable load forecasting provides indispensable guidance for optimal unit commitment, efficient power distribution, and energy efficiency and thus plays a crucial role in power systems. Recently, the Energy Internet (EI) [1], as a further upgrade of the smart grid, has attracted considerable attention from academia and industry. The EI not only innovates the infrastructure, but also puts higher demands on the intelligent management of energy [2]. On the one hand, due to the high penetration of plug-and-play renewable energy resources in the EI, their variable output will exacerbate grid load volatility; on the other hand, the EI transforms the centralized infrastructure into distributed energy management, in which the distributed microgrids with different scales are responsible for managing densely deployed energy supply/demand units [3,4]. Fully exploiting the spatiotemporal correlation of load trends between geographically close microgrids will greatly facilitate forecasting tasks. Therefore, load forecasting faces new challenges and opportunities against the backdrop of the EI.

Various load forecasting techniques have emerged over the past decades [5–9], among which artificial intelligence based methods have become promising solutions because they excel at mapping the relationship between dependent and independent variables [10,11]. Reis et al. [6] embedded the discrete wavelet transform (WT) into the multilayer perceptron and proposed a multi-model short term load forecasting scheme. In [7], the authors employed support vector regression (SVR) machines to build a parallel forecasting architecture for hourly load in a day and used particle swarm search to optimize the SVR hyperparameters. Lin et al. [8] proposed an ensemble model based on variational mode decomposition (VMD) and extreme learning machine (ELM) for multi-step ahead load forecasting. Ko et al. [12] developed a model based on SVR and radial basis function networks and leveraged the extended Kalman algorithm to filter the model parameters. In addition, the ELM [7,13], bi-square kernel regression model [14], and k-nearest neighbor algorithm [15] were employed to fulfill forecasting tasks. However, the above traditional machine learning methods and shallow networks are inadequate to fully model the complexity of the power demand side and thus often suffer from limited accuracy or stability.

Over the past decade, deep learning (DL) has achieved great success in many fields [16,17]. DL can spontaneously extract valuable information from a variety of related factors and then leverage their powerful nonlinear representation ability to pursue ideal results. Inspired by this, some popular DL based models, such as the feed-forward deep neural network (FF-DNN) [18], convolutional neural network (CNN) [19], deep recurrent neural network (RNN) [20], and long short term memory (LSTM) [21,22], have been applied to load forecasting and shown excellent performance.

In [23], the authors evaluated the effects of deep stacked autoencoders and RNN-LSTM forecasting models, and the test results suggested the superiority of the DL based method compared to traditional models. In [18], the authors built models using FF-DNN and deep RNN and extracted up to 13 features from the raw load and meteorological data to drive the model. Li et al. [19] employed deep CNN to cluster two-dimensional input loads, then combined it with various weather variables and a feed-forward network to obtain better forecast results. Shi et al. [20] developed a deep recurrent architecture using the LSTM unit and merged residential load and associated weather information as input data. In recent studies, the work in work [22] used the density based clustering technique to preprocess historical information such as loads and holidays and built an LSTM based model that can process multiple time step data simultaneously; The Copula function and deep belief network were used in [24] to establish an hour-ahead forecasting method, where electricity prices and temperatures were introduced into the input; Farfar et al. [25] proposed a two stage forecasting system, in which the k-means first clusters the forecasted load according to the estimated temperature, then multiple stacked denoising autoencoder (SDAE) based models for different load clusters perform prediction in the second stage; In [26], cutting edge deep residual networks were modified into a forecasting model and combined with Monte Carlo dropout to achieve probabilistic forecasting.

The above mentioned deep models hope to rely on redundant connections to accommodate as many fluctuation patterns as possible, thereby improving the inference robustness. Unfortunately, this vision is difficult to achieve for complex and volatile electricity consumption problems, and these DL based methods often struggle with performance degradation elicited by two major problems: (1) The constructed input data usually fail to exploit historical load information fully. Therefore, to achieve the desired result, they have to resort to more related variables (e.g., over 10 types of variables in [18,26]), along with the necessary data preprocessing. (2) The developed deep framework usually focuses on creating precise mapping from input to output, which may be unreliable or not even hold in practice, which makes them less flexible and self-regulating in the face of increasingly complex and diverse load situations [11]. To improve the forecasting performance, some methods [20,25] have to train separate models according to different load data patterns.

The major challenge of load forecasting technology is how to meet the diversity of data patterns brought by strong randomness and volatility. A daily load curve typically consists of (1) the cyclical

load (accounting for a relatively large proportion) in the regular pattern, (2) the uncertain load (small proportion) caused by external factors such as weather, holidays, and customer behavior, and (3) the noise that cannot be physically explained (minimum proportion) [20], all of which are ultimately quantified and superimposed as a load sequence. Sufficient load sequences represent a variety of data patterns, and thus, reliable forecasts can be achieved by mining large amounts of highly correlated load data only, which has also been confirmed in several recent studies [9,20,22]. However, since the uncertainty and noise account for less in the load, the contribution to the total forecast error is relatively small, which makes it easier for the forecasting model to remember the data pattern of cyclical load, but not the fluctuation. As a result, many models, on the one hand, need to resort to more types of input variables to remedy the neglect of the fluctuation data patterns due to poor representation learning or nonlinear mapping capabilities; on the other hand, they have to face the complex data preprocessing and cumulative errors resulting from multivariate inputs.

In this work, we propose an ultra short term (one-hour-ahead) load forecasting scheme to provide decision support for power quality control, online operation safety monitoring and prevention, and emergency control. The scheme designs an input data plan based on historical load only and develops a deep model with excellent representation learning and regression capabilities, aiming to address the problems mentioned above as much as possible.

Specifically, we first formulated a historical load matrix (HLM) in the context of the EI, which covers load data for multiple zones at different time points. Second, we leveraged the HLMs to create a historical load tensor (HLT), aiming to provide a forecasting model with enough source materials. Third, based on the spatiotemporal correlation of our HLM, we proposed a novel matrix decomposition algorithm to separate the base load and random components in the HLT effectively. Finally, we calculated the gradient information of the HLTs and formed all the HLT based preprocessing results into a multidimensional array to drive the forecasting model.

Further, to transform the constructed input into the desired forecast results, we developed a forecasting model consisting of a feature learning module and a regression module. The feature learning module is based on the 3D CNN [27] architecture and can extract valuable data from three input dimensions (i.e., depth, width, and height), which greatly improves the richness of learned features. For the regression module, given the consecutive time attribute of the learned feature sequences, we implemented nonlinear mapping, employing a gated recurrent unit (GRU) [28] that works well on time series based tasks. The GRU is in the family of recurrent networks and is as good at tackling cases with long term dependencies as LSTM, but with much fewer network parameters [29,30]. The proposed model, the 3D CNN-GRU, closely matches the constructed input data and can spontaneously explore various data patterns from the fed multidimensional arrays, thereby facilitating forecasting performance.

Figure 1 presents an overview of our load forecasting scheme against the backdrop of the EI. This work mainly contributes three points: (1) In the EI context, we created an HLT with spatiotemporal correlation and then preprocessed it based on matrix low rank decomposition and the load gradient to get refined hierarchical input data. (2) We developed a novel 3D CNN-GRU model, which consists of two functional modules and can forecast the load trend of any zone covered by the HLM by changing the load label. (3) We constructed the input data based on real-world data and built the 3D CNN-GRU model with TensorFlow [31]. Self-assessments from multiple perspectives and comprehensive comparisons with several advanced methods were carried out.

The rest of this paper is organized as follows. Section 2 details the proposed forecasting scheme. The experimental evaluations and analyses are in Section 3. In Section 4, the conclusions and prospect are given.

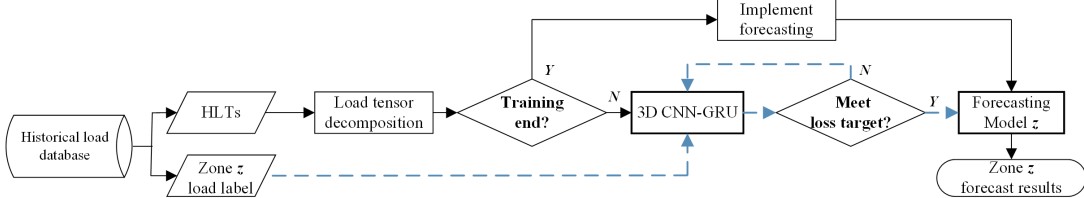

**Figure 1.** Flowchart of the proposed scheme. The "zone *z*" is one of the zones covered by the historical load matrix (HLM).

## 2. Proposed Solution

In this section, we detail the input data construction and forecasting model framework design.

### 2.1. Model Input Data

According to the existing research [32–34], the daily load curves for several consecutive days in a given zone have consistent trends due to similar external factors such as weather, human behavior, and infrastructure. Therefore, daily load ahead of the forecasted hour for several consecutive days must contain a large amount of information for the inference. In addition, due to factors such as a similar climate and holidays, geographically close zones should have certain similarities in load variability [35]. Consequently, for a given forecast zone, making full use of spatiotemporally related information will be beneficial for forecasting tasks.

#### 2.1.1. Historical Load Tensor

Let *t* denote a recording time point of the historical load, then our formulated HLM at hour *t* can be expressed as:

$$\boldsymbol{L}^t = \begin{pmatrix} \boldsymbol{l}_1^t \\ \boldsymbol{l}_2^t \\ \vdots \\ \boldsymbol{l}_D^t \end{pmatrix} = \begin{pmatrix} l_{1,1}^t & l_{1,2}^t & \cdots & l_{1,Z}^t \\ l_{2,1}^t & l_{2,2}^t & \cdots & l_{2,Z}^t \\ \vdots & \vdots & \ddots & \vdots \\ l_{D,1}^t & l_{D,2}^t & \cdots & l_{D,Z}^t \end{pmatrix} = \left( l_{d,z}^t \right) \in \mathbb{R}^{D \times Z}, \quad \forall t \in \{0, 1, ..., 23\}. \tag{1}$$

Here, $\boldsymbol{L}^t$ consists of multi-zone load vectors, and $(\boldsymbol{l}_1^t, ..., \boldsymbol{l}_D^t)^T$ represents the load at hour *t* for consecutive *D* days. The selected days need be close to the forecasted time. The HLM not only provides more spatiotemporal information, but can (*D* is large enough) indirectly incorporate the holiday factor. The multi-zone load vector $\boldsymbol{l}_d^t = (l_{d,1}^t, l_{d,2}^t, \ldots, l_{d,Z}^t)$ covers the load data of different zones at the same hour *t*, and $l_{d,z}^t$ denotes the load value of zone *z* on day *d*.

In this work, we selected 12 geographically close zones from the PJM [36] to represent distributed microgrids in the EI: AECO, BC, DPLCO, DUQ, JC, ME, PAPWR, PE, PEPCO, PS, RECO, and UGI. PJM is a regional transmission organization in the U.S. that currently operates an electric transmission system serving all or parts of the 13 eastern states and Washington, D.C., covering 29 electricity zones including the above 12 [37]. Although the load aggregation scales in different zones are inconsistent, we focus on inferring the load curve trend, and all raw values need be quantified; it thus has a small impact on the theoretical evaluation of the proposed forecasting scheme. In addition, load aggregation at different scales is more in line with the characteristics of distributed units and can better reflect the generalization of the proposed model, which will be demonstrated in the experimental results.

Due to the close geographic distance and the same time, there will be a strong spatiotemporal correlation between the HLM elements [32], which is a prerequisite for the matrix to have (approximately) low rank and will be demonstrated in the next subsection.

Let $t + 1$ indicate the forecasted hour, and the previous load information of all zones is known. With the HLMs, the HLT used to infer the load of zone $z$ at hour $t + 1$ can be expressed as:

$$\boldsymbol{\mathcal{T}}^t = \left( (\boldsymbol{L}^t), (\boldsymbol{L}^{t-1}), \ldots, (\boldsymbol{L}^{t-n}), \ldots, (\boldsymbol{L}^{t-(N-1)}) \right) \in \mathbb{R}^{N \times D \times Z}, \quad 1 \le N \le 23, \tag{2}$$

where $t$ to $t - (N - 1)$ denote $N$ consecutive hours ahead of the forecasted time point $t + 1$. In the tensor $\boldsymbol{\mathcal{T}}^t$, we call herein each $\boldsymbol{L}^{t-n}$ a "frame", i.e., the HLT is composed of $N$ HLM frames. Note here the natural day based periodicity of variable $t$.

### 2.1.2. Load Tensor Decomposition

We first demonstrate that the HLM has low rank or approximately low rank. Figure 2a visualizes the trend of the annual load profiles for all 12 zones, i.e., spatial correlation; Figure 2b shows the trend of the annual load for one given zone at different hours, i.e., temporal correlation. Further, we theoretically demonstrate the low rank property of the HLM via singular value decomposition (SVD).

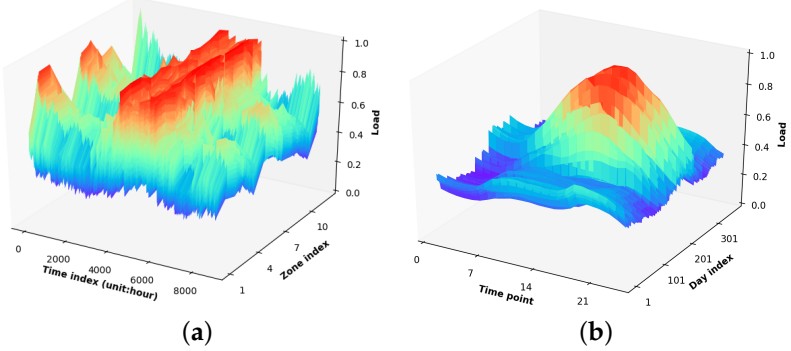

**Figure 2.** Illustration for the load profiles. The data are from (**a**) 12 zones and (**b**) the RECO, respectively.

In Figure 3, the first few singular values (SVs) occupy most of the energy of the HLM, while the extra SVs resulting from the fluctuating load are very small. These indicate the low rank property of the HLM and support the feasibility of conducting matrix low rank decomposition.

For an HLM $\boldsymbol{L}^t$ with (approximately) low rank, it can be expressed as $\boldsymbol{L}^t = \boldsymbol{X} + \boldsymbol{E}$ according to the low rank decomposition theory [38], where $\boldsymbol{X}$ and $\boldsymbol{E}$ denote a very low rank matrix and a noise matrix, respectively. Solving $\boldsymbol{X}$ and $\boldsymbol{E}$ can be equivalent to a rank minimization problem, as follows:

$$\min_{\boldsymbol{X}} \ \text{rank}(\boldsymbol{X}) \qquad \text{s.t.} \ \boldsymbol{L} = \boldsymbol{X} + \boldsymbol{E}, \quad \boldsymbol{L}, \boldsymbol{X}, \boldsymbol{E} \in \mathbb{R}^{D \times Z}. \tag{3}$$

Here, $\boldsymbol{L}$ denotes the HLM $\boldsymbol{L}^t$ for simplicity.

In our cases, $\boldsymbol{X}$ stands for the base load matrix (BLM), which refers to the expectation of cyclical loads caused by relatively stable daily behavior. The noise $\boldsymbol{E}$ represents the fluctuation load matrix (FLM), which reflects the load generated by various random phenomena [34].

Based on previous studies [38,39], the NP-hard problem (3) can be relaxed by the convex problem:

$$\min_{\boldsymbol{X}} \ \|\boldsymbol{X}\|_* + \lambda \|\boldsymbol{E}\|_{2,1} \qquad \text{s.t.} \ \boldsymbol{L} = \boldsymbol{X} + \boldsymbol{E}, \tag{4}$$

where $\|E\|_{2,1} = \sum_{z=1}^{Z} (\sum_{d=1}^{D} ((E)_{d,z})^2)^{1/2}$ is the $\ell_{2,1}$ norm, which is used to encourage column vectors' sparsification. Taking the basic load as the baseline, the $\ell_{2,1}$ norm is physically meaningful because the short term daily load of a zone at the same hour often fluctuates similarly (see Figure 2b). The hyper parameter $\lambda$ is used to fine tune the fluctuations ratio, whose typical range is 0.4~0.6 in our cases.

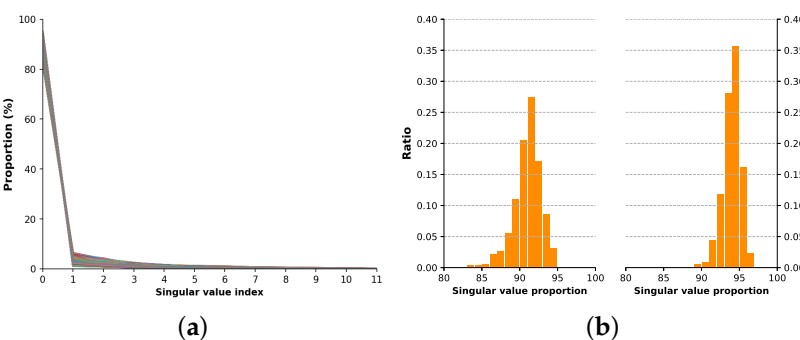

**(a)**　　　　　　　　　　　　　　　　　　　　**(b)**

**Figure 3.** Proportion of the historical load matrix (HLM) singular values (SVs). The HLMs stem from 2018 and are $14 \times 12$ in size. (**a**) Proportion of each SV; (**b**) proportion of (left) the first SV and (right) the first two.

The numerical iterative algorithm is the dominant solution to Problem (4). Based on the existing research and the precision requirements for decomposition in our scenario, we propose a distributed augmented Lagrange multipliers (DALM) algorithm based on the alternating direction method of multipliers (ADMM) [40] framework to achieve optimization effectively.

Firstly, the Lagrange function of Problem (4) can be given by:

$$\mathcal{L}(X, E, Y, \rho) = \|X\|_* + \lambda \|E\|_{2,1} - \langle Y, L - X - E \rangle_F + \frac{\rho}{2} \|L - X - E\|_F^2, \tag{5}$$

where $Y$ and $\rho$ are the multiplier matrix and penalty factor, respectively.

Then, we make two transformations for (5), as follows:

$$\begin{aligned} -\langle Y, L - X - E \rangle_F + \frac{\rho}{2} \|L - X - E\|_F^2 &= tr\{-(L - X - E)^{\mathrm{T}} Y + \frac{\rho}{2} (L - X - E)^{\mathrm{T}} (L - X - E)\} \\ &= \frac{\rho}{2} \|L - X - E - \rho^{-1} Y\|_F^2 - (2\rho)^{-1} \|Y\|_F^2. \end{aligned} \tag{6}$$

Finally, the Lagrange function (5) can be expressed as:

$$\mathcal{L}(X, E, Y, \rho) = \|X\|_* + \lambda \|E\|_{2,1} + \frac{\rho}{2} \|L - X - E - \rho^{-1} Y\|_F^2 - (2\rho)^{-1} \|Y\|_F^2. \tag{7}$$

With the ADMM, the specific variables update procedure of Problem (7) as follows.

(1) Fix the remaining parameters, and update $E$.

$$\begin{aligned} \mathcal{L}(E) = \lambda \|E\|_{2,1} + \frac{\rho}{2} \|L - X - E - \rho^{-1} Y\|_F^2 &= \lambda \sum_{z=1}^{Z} (\sum_{d=1}^{D} ((E)_{d,z})^2)^{1/2} + \frac{\rho}{2} \sum_{z=1}^{Z} \sum_{d=1}^{D} ((L - X - E - \rho^{-1} Y)_{d,z})^2 \\ &\leq \lambda \sum_{z=1}^{Z} (\sum_{d=1}^{D} |(E)_{d,z}|) + \frac{\rho}{2} \sum_{z=1}^{Z} \sum_{d=1}^{D} ((E - (L - X - \rho^{-1} Y))_{d,z})^2 \overset{\text{def}}{=} \Theta(E). \end{aligned} \tag{8}$$

We define herein a function:

$$\theta(e_{d,z}) = \lambda \left| e_{d,z} \right| + \frac{\rho}{2}(e_{d,z} - m_{d,z})^2, \quad \forall d \in \{0,1,...,D\}, \forall z \in \{0,1,...,Z\}, \tag{9}$$

where $\forall e_{d,z} \in \{(E)_{d,z}\}$ and $\forall m_{d,z} \in \{(L - X - \rho^{-1}Y)_{d,z}\}$. Apparently, there are $\Theta(E) = \sum_{d,z} \theta(e_{d,z})$. Further, we calculate the partial derivative of $\theta(e_{d,z})$ with respect to variable $e_{d,z}$, as follows:

$$\nabla_{e_{d,z}} \theta(e_{d,z}) = \lambda \text{sgn}(e_{d,z}) + \rho(e_{d,z} - m_{d,z}), \tag{10}$$

where $\text{sgn}(\cdot)$ is the Signum function. With Equation (10), the $\theta(e_{d,z})$ minimum can be given by:

$$\arg\min_{e_{d,z}} \theta(e_{d,z}) = \begin{cases} m_{d,z} - \lambda/\rho & m_{d,z} \geq \lambda/\rho \\ 0 & |m_{d,z}| < \lambda/\rho \\ m_{d,z} + \lambda/\rho & m_{d,z} \leq -\lambda/\rho \end{cases}. \tag{11}$$

Introduce a shrinkage operator $\mathcal{S}_\tau(x) \stackrel{\text{def}}{=} \text{sgn}(x)[|x| - \tau]_+$, where $[\cdot]_+ = \max(\cdot, 0)$. Then, Equation (11) can be expressed as:

$$\arg\min_{e_{d,z}} \theta(e_{d,z}) = \text{sgn}(m_{d,z})[|m_{d,z}| - \lambda/\rho]_+ = \mathcal{S}_{\lambda/\rho}(m_{d,z}). \tag{12}$$

Ultimately, the optimal $E^*$ can be given by:

$$E^* = \arg\min_E \Theta(E) = \arg\min_{e_{d,z}} \sum_{d,z} \theta(e_{d,z}) = \mathcal{S}_{\lambda/\rho}(L - X - \rho^{-1}Y). \tag{13}$$

Accordingly, update $E$ in this step can be equivalent to:

$$E^k := \mathcal{S}_{\lambda/\rho}(L - X^{k-1} - \rho^{-1}Y^{k-1}). \tag{14}$$

(2) Fix $E^k$ and other parameters, and update $X$.

$$\mathcal{L}(X) = \|X\|_* + \frac{\rho}{2} \|L - X - E - \rho^{-1}Y\|_F^2. \tag{15}$$

The optimal $X^*$ is expressed as:

$$X^* = \arg\min_X \rho^{-1}\|X\|_* + \frac{1}{2}\|X - (L - E - \rho^{-1}Y)\|_F^2 = \mathcal{D}_{1/\rho}(L - E - \rho^{-1}Y), \tag{16}$$

where the $\mathcal{D}_{1/\rho}(\cdot)$ represents a singular value shrinkage operator [39], defined as:

$$\mathcal{D}_\tau(M) = U\mathcal{S}_\tau(\Sigma)V^T, \quad \mathcal{S}_\tau(\Sigma) = \begin{cases} \sigma_i - \tau & \sigma_i > \tau \\ 0 & 0 \leq \sigma_i \leq \tau \end{cases}, \tag{17}$$

where $\Sigma$, $U$, and $V$ are the diagonal matrix of singular values $\sigma_i$ and the left and right orthogonal matrices of $M$, respectively. Therefore, update $X$ can be expressed as:

$$X^k := \mathcal{D}_{1/\rho}(L - E^k - \rho^{-1}Y^{k-1}). \tag{18}$$

(3) Fix the others, and update multiplier matrix $Y$.

$$\mathcal{L}(Y) = \frac{\rho}{2}\|L - X - E - \rho^{-1}Y\|_F^2 - (2\rho)^{-1}\|Y\|_F^2. \tag{19}$$

Obviously, Equation (19) is strongly convex, and thus, its minimum value can be found by the partial derivative as follows:

$$\nabla_Y \mathcal{L} = \frac{\partial}{\partial Y}tr\{\frac{\rho}{2}(L - X - E - \rho^{-1}Y)^T(L - X - E - \rho^{-1}Y) - (2\rho)^{-1}Y^T Y\} = -(L - X - E). \tag{20}$$

Update $Y$:

$$Y^k := Y^{k-1} - \rho(L - X^k - E^k). \tag{21}$$

(4) Update penalty factor $\rho$.

$$\rho^k := \alpha\rho^{k-1}, \quad \alpha > 1. \tag{22}$$

The above procedure is outlined in Algorithm 1. With the ADMM, our optimization problem is distributed into four convex objectives, which guarantees the convergence and stability of the final solution. The low rank decomposition can effectively separate cyclical load and fluctuation components in the HLT, which greatly facilitates the feature learning of the 3D CNN. The performance gains brought by the DALM will be shown later.

---

**Algorithm 1** Distributed augmented Lagrange multipliers algorithm.

---

**Input:** Load matrix $L \in \mathbb{R}^{D \times Z}$, the parameters $\lambda$, $K_{\max}$, $Y^0 = 0$, $\rho^0 = 10^{-5}$, $\alpha = 1.1$, $\epsilon = 10^{-5}$.
**Output:** Optimal matrices $X^*$ and $E^*$.
  1: Initialize the basic load matrix $X^0$.
  2: **for** $k = 1, 2, ..., K_{\max}$ **do**
  3:     $i \leftarrow 0$; $X_i^* \leftarrow X^{k-1}$
  4:     **while** False **do**
  5:         $E^k \leftarrow \mathcal{S}_{\lambda/\rho}(L - X_i^* - \rho^{-1}Y^{k-1})$
  6:         $\triangleright$ Perform SVD.
  7:         $U, \Sigma, V \leftarrow \text{SVD}(L - E^k - \rho^{-1}Y^{k-1})$; $X_{i+1}^* \leftarrow U\mathcal{S}_{1/\rho}(\Sigma)V^T$
  8:         $\triangleright$ Check convergence.
  9:         **if** $\|X_{i+1}^* - X_i^*\|_F^2 \leq \epsilon$ **then**
10:            $X^k \leftarrow X_i^*$
11:            **break**
12:         **end if**
13:         $i+ = 1$
14:     **end while**
15:     $\triangleright$ Check convergence.
16:     **if** $\|X^k - X^{k-1}\|_F^2 \leq \epsilon$ **then**
17:         **break**
18:     **end if**
19:     Update $Y^k$ by Equation (21)
20:     Update $\rho^k$ by Equation (22)
21: **end for**
22: **return** $X^* = X^k$, $E^* = E^k$.

---

### 2.1.3. Load Gradient

To understand the load variation better, we provide two load gradients with different time scales. We first calculate the frame gradient of the HLT by backward difference, as follows:

$$
\boldsymbol{L}_{\nabla t}^{t} = \boldsymbol{L}^{t} - \boldsymbol{L}^{t-1} = \begin{pmatrix} \boldsymbol{l}_1^t - \boldsymbol{l}_1^{t-1} \\ \vdots \\ \boldsymbol{l}_D^t - \boldsymbol{l}_D^{t-1} \end{pmatrix} = \left( l_{d,z}^{\nabla t} \right) \in \mathbb{R}^{D \times Z}. \tag{23}
$$

$\boldsymbol{L}_{\nabla t}^{t}$ denotes a frame gradient matrix (FGM), which reflects the load variations with the recording hour.

The day gradient matrix (DGM) is used to describe the change information of adjacent daily loads at the same hour, expressed as:

$$
\boldsymbol{L}_{\nabla d}^{t} = \begin{pmatrix} \boldsymbol{l}_1^t - \boldsymbol{l}_0^t \\ \boldsymbol{l}_2^t - \boldsymbol{l}_1^t \\ \vdots \\ \boldsymbol{l}_D^t - \boldsymbol{l}_{D-1}^t \end{pmatrix} = \begin{pmatrix} l_{1,1}^t - l_{0,1}^t & \cdots & l_{1,Z}^t - l_{0,Z}^t \\ l_{2,1}^t - l_{1,1}^t & \cdots & l_{2,Z}^t - l_{1,Z}^t \\ \vdots & \ddots & \vdots \\ l_{D,1}^t - l_{D-1,1}^t & \cdots & l_{D,Z}^t - l_{D-1,Z}^t \end{pmatrix} = \left( l_{\nabla d,z}^t \right) \in \mathbb{R}^{D \times Z}, \tag{24}
$$

where $\boldsymbol{l}_0^t$ represents the previous daily load vector next to the start day of $\boldsymbol{L}^t$.

Summarized, the input data fed to the forecasting model can eventually be expressed as:

$$
\boldsymbol{\mathcal{X}}^t = \left( ..., ((\boldsymbol{X}^{t-n}), (\boldsymbol{E}^{t-n}), (\boldsymbol{L}_{\nabla t}^{t-n}), (\boldsymbol{L}_{\nabla d}^{t-n})), ... \right) \in \mathbb{R}^{N \times D \times Z \times 4}, \quad \forall n \in \{0, 1, ..., N-1\}. \tag{25}
$$

Finally, since our model training performs a supervised learning, an input $\boldsymbol{\mathcal{X}}^t$ for training needs to correspond to an actual load label. We represent the label dataset as $\{l_{z,i}^{t+1}\}$, where $z$ and $t+1$ denote the forecasted zone and hour, respectively.

### 2.2. 3D CNN-GRU

In this subsection, we detail the developed forecasting model and training process. The representative model framework is shown in Figure 4.

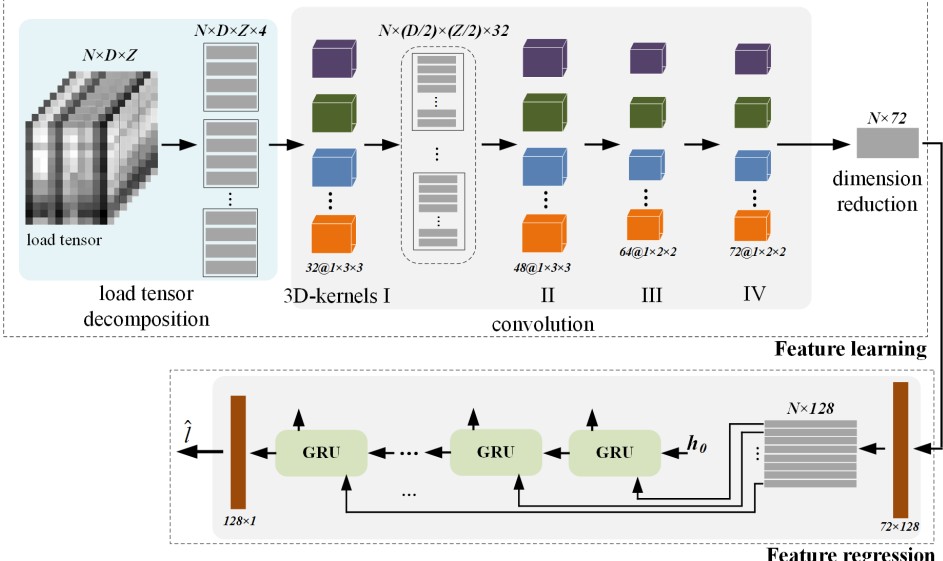

**Figure 4.** Illustration of the proposed forecasting model. In the feature learning module, the convolution stride of the frame direction is set to one, and the height and width direction are two, while the padding is "SAME". Activation function selects the rectified linear unit (ReLU). The employed classical GRU [30] has a hidden layer of 128. Note that a typical size of load tensor matching this framework is $8 \times 14 \times 12$, where "8" indicates that the load values at eight consecutive time points before the forecasted hour are adopted and "14" means that each time point contains data for fourteen consecutive days.

### 2.2.1. Learning Module

The learning module is used to extract features from the input load data, which is achieved through three segments in turn. The first step is data preprocessing, where each frame of the HLT is decomposed into four channels via the DALM and gradient operations and fed into the 3D CNN. The second employs several sets of 3D convolutions to achieve different levels of feature learning. Taking the framework in Figure 4 as an example, the output after the first convolution can be expressed as:

$$\boldsymbol{\mathcal{Y}}_{n,j}^{o_1} = \text{ReLU}((\sum_{i=0}^{3} \boldsymbol{\mathcal{X}}_{n,:,:,i}^{t} \otimes \boldsymbol{\mathcal{W}}_{1,i,j}^{l_1}) + \boldsymbol{b}_{j}^{l_1}), \quad \forall j = \{0,1,...,31\}, \tag{26}$$

where $\boldsymbol{\mathcal{Y}}_{n,j}^{o_1}$ represents the output feature of the $n^{\text{th}}$ frame in channel $j$. Since the frame stride and depth of the 3D convolution are both one, the learned features from each HLM maintain a high independence, which benefits the GRU based regression network.

The first layer focuses on extracting implicit low level features that reflect the profiles of input load data, and the following convolutions are operationally identical to the first. Note that the size of the feature matrix learned is halved compared to the input. Thus, to extract information effectively and reduce the null operation, the kernel size of the back-end convolutions should be reduced.

At the convolution end, we simplify the output array of $N \times 1 \times 1 \times 72$ into a feature matrix and feed it to the regression module. Due to the learning pattern we adopt, each row of the learned matrix is a feature sequence originating from each frame, which maintains a relative independence of the load features over time and better matches the operating mechanism of the GRU.

## 2.2.2. Regression Module

The regression module is mainly composed of $N$ classic GRU cells, in which we still use the ReLU activation function in the input and output layers.

When performing regression, the input layer first re-extracts the features of input data to form an $N \times 128$ array. Then, each row of the input array is fed as a vector to the corresponding GRU cells, which have 128 nodes and employ the same internal structure as [30]. Last, the output layer processes the $N^{\text{th}}$ GRU output using 128 single layer nodes to obtain the forecast load of a given zone.

## 2.2.3. Training

During training, we select the input data $\{\boldsymbol{\mathcal{X}}^t\}$ and the label set corresponding to the forecasted zone and feed it to the model until convergence. As a regression problem, the loss function of the 3D CNN-GRU is the mean square error (MSE), as follows:

$$L_{loss} = \frac{1}{2} \sum_i \left( l_{z,i}^{t+1} - \hat{l}_i \right)^2, \tag{27}$$

where $\hat{l}_i$ denotes the forecast load. By selecting different label data, we can train different models for the load forecasting of the corresponding zone.

## 3. Experiments and Analysis

In this section, we first describe the experimental dataset and comparison models. Second, we evaluate and analyze the proposal from different perspectives. In the final subsection, we present a comparison with several state-of-the-art (SOTA) forecasting methods.

### 3.1. Data Description

The real-world data came from 12 geographically close zones [37] of the PJM [36], with a time span of December 2011 to June 2019 and a resolution of one hour. We selected the PAPWR, RECO, and UGI, whose load ranges (in MW) were [204.4, 1,053.8], [83.1, 438.4], and [64.7, 223.5], respectively, as three study cases (i.e., the evaluation of three corresponding models). The load label set of each zone covered the period from January 2012 to June 2019, totaling 65,712 (2738 days $\times$ 24 h), and the number of input data was the same as the label. Note that we here used some data from December 2011.

In the constructed input data, we picked out the last seven daily loads from the monthly data in 2017 and 2018 for testing, totaling 4032 per zone. In addition, the 2019 data were used for testing only. The data of each zone were normalized with reference to their own maximum load.

It needs to be stated that the load was a time series, and using a model trained by 2018 data to test the 2017 case was not practical. However, given the similarity of load profiles year-on-year, we ignored the time attribute of the case and instead focused on the data diversity represented by the samples.

### 3.2. Comparison Methods

We selected several SOTA methods for comparison: SSA-SVR [7], ELM-MABC [13], KMC-SDAE [25], and DBC-LSTM [22]. For fairness, all methods used the same raw data and time resolution, and the weather information was from an open platform [41].

Specifically, for the SSA-SVR, we selected weather information from Pittsburgh and New York to generate the meteorological variables for the PAPWR and RECO (the same below); an ELM-MABC of 20 hidden nodes performed forecasting tasks, and only one-hour-ahead forecasting was conducted. For the KMC-SDAE, we provided the model with real hourly temperature data instead of estimations, and the

amount of training data was consistent with ours. For the DBC-LSTM, only the best model with 12 time steps was adopted, and the training data range was extended to 12 months.

In addition, we compared with two traditional autoregressive (AR) models, knmV-AR [32] and NCST-LF [33], which also explored the load spatiotemporal correlation, in order to suggest the advantages of the proposed scheme in the utilization of spatiotemporal data. For the two AR models, we replaced their meter measured data with zones' data and used the columns of the created HLM as the historical load vector.

### 3.3. Performance Evaluation

According to the foregoing, the three forecasting models were trained separately with the frame in Figure 4 and the HLT of $8 \times 14 \times 12$. We first target the PAPWR and show the overall forecasting effect.

Figure 5 shows the seven day forecast curves of the proposed scheme for different months. It can be found that our proposal had good overall performance for different forecasting periods, and the inferred results basically matched the actual load curves. The reasons were chiefly as follows: (1) The created load tensor covering multiple zones and time points provided more valuable information for the 3D CNN-GRU, and the proposed preprocessing methods, especially the low rank based matrix decomposition, emphasized the characteristics of the load data, thereby cementing the pertinence of feature learning. (2) Benefiting from the hierarchical convolutions in three directions, the 3D CNN was favorable in terms of the feature learning of a multidimensional array. It not only ensured temporal continuity of the learned feature sequences, but maintained independence between frames. (3) While our scheme did not take into account related variables such as weather, the input data covered multiple days and zones and incorporated as many potential factors that affect future loads as possible; meanwhile, the trained model was capable of spontaneously mining the desired features and conducting powerful nonlinear mapping, which contributed to the improvement of prediction performance.

Besides the above, the magnitude of daily load variance also affected the accuracy and stability. In detail, for relatively stable cases, e.g., Figure 5b,c, the proposal had better results. The reason was that the periodic base load was dominant in such daily curves; thus, the decomposed BLMs had far more energy than the random components, while it was easier for the 3D CNN to remember the data pattern of strong features. By contrast, for the period when electricity consumption fluctuated greatly, e.g., Figure 5a,d, although the separated FLM facilitated the model to learn the refinement features of fluctuations, the strong randomness made the regression module unable to be a perfect mapper.

Then, we introduced three common criteria, mean absolute error (MAE), root mean squared error (RMSE), and mean absolute percentage error (MAPE), to quantify the model performance, as follows:

$$\text{MAE} = \frac{1}{N_s} \sum_{i=1}^{N_s} |l_i - \hat{l}_i|, \quad \text{RMSE} = \sqrt{\frac{1}{N_s} \sum_{i=1}^{N_s} \left(l_i - \hat{l}_i\right)^2}, \quad \text{MAPE} = \frac{100}{N_s} \sum_{i=1}^{N_s} \frac{|l_i - \hat{l}_i|}{l_i}, \tag{28}$$

where $l_i$ and $\hat{l}_i$ stand for the actual value and the forecast result, respectively. In the test, $N_s$ was set according to different statistical ranges.

We calculated two year test errors for two zones, and the statistical range of the samples was one day, i.e., $N_s = 24$. Figure 6 shows the quantified performance of the model represented by the MAE and RMSE. For the RECO with a larger capacity, over 90% of the MAE was lower than 10 MW, in which nearly half of the results were less than 5 MW; while Figure 6a also visually suggests the stability of ours, that is the RMSE curve generally deviated from MAE slightly. Similarly, the above situations occur in the forecasts of Figure 6b.

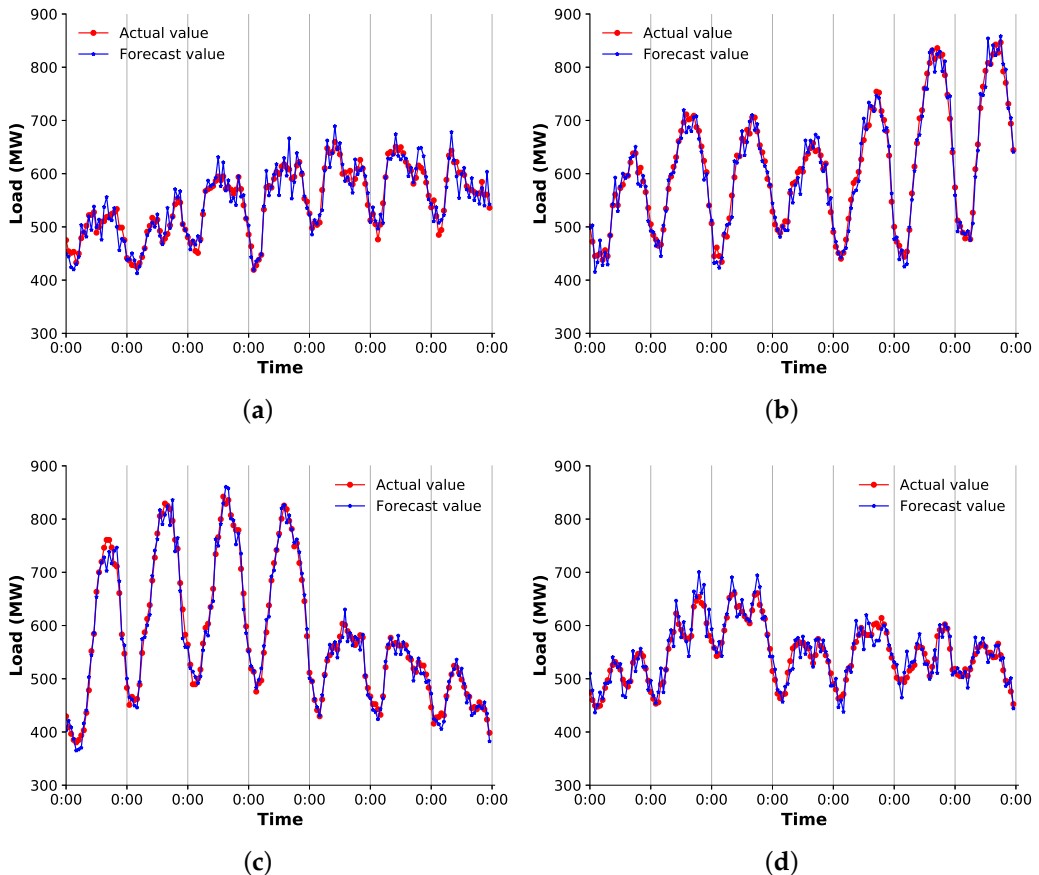

**Figure 5.** Forecast results at different times. The raw data are from the PAPWR, and the time is (**a**) March 2017, (**b**) June 2018, (**c**) September 2017, and (**d**) December 2018, respectively. Note that the original forecast values are denormalized, and "0:00" indicates the starting hour of a natural day.

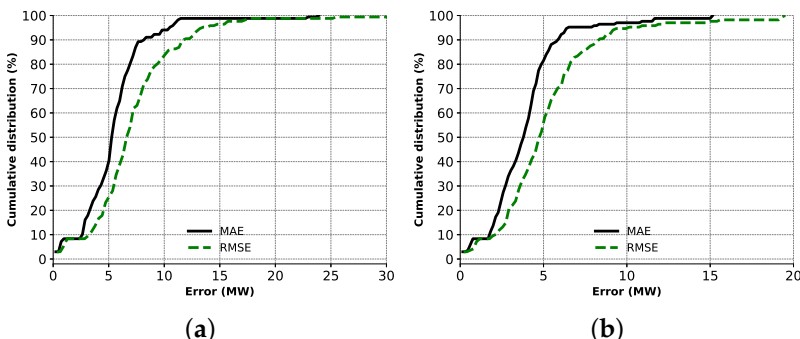

**Figure 6.** Error distribution of the forecast results. The data are from the (**a**) RECO and (**b**) UGI.

Finally, we assessed the proposal with seven day data from 2019, and Figure 7 draws the forecasted profiles. For the absolute forecasting period (in which no training data were provided to the model), the proposed model still achieved a good result, which suggested that it had certain applicability. The main reasons were the spontaneous feature mining ability and the favorable nonlinear representation of the 3D CNN-GRU, along with the constructed multidimensional input.

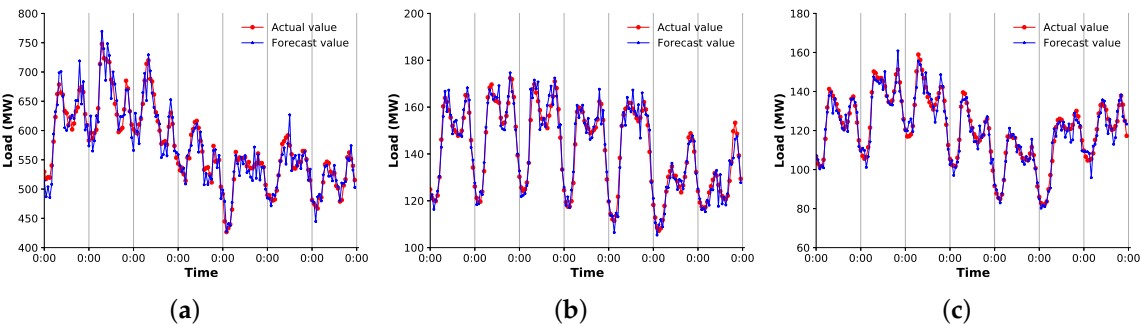

**Figure 7.** Forecast results for different zones. The data from (**a**) PAPWR, (**b**) RECO, and (**c**) UGI in March 2019.

Furthermore, to test the influence of diverse input data patterns and different model structures on forecast accuracy and stability, we separately evaluated the proposal from three aspects.

First, we fixed the frame number and created three additional HLMs covering different days $D$ and then trained and tested the model with the PAPER. Note that for different matrix sizes, the convolutional layer of 3D CNN needed to be adjusted (see Table 1). Moreover, the MAPE was calculated by monthly sample ($N_s = 168$).

**Table 1.** Convolution settings for different HLMs.

| Filter Bank<br>HLM Size | I | II | II | IV | V |
|---|---|---|---|---|---|
| $7 \times 12$ | $32@1 \times 2 \times 2$ | $48@1 \times 2 \times 2$ | $64@1 \times 2 \times 2$ | $72@1 \times 1 \times 2$ | - |
| $14 \times 12$ | $32@1 \times 3 \times 3$ | $48@1 \times 3 \times 3$ | $64@1 \times 2 \times 2$ | $72@1 \times 2 \times 2$ | - |
| $21 \times 12$ | $32@1 \times 3 \times 3$ | $48@1 \times 3 \times 3$ | $64@1 \times 2 \times 2$ | $64@1 \times 2 \times 2$ | $72@1 \times 2 \times 1$ |
| $28 \times 12$ | $32@1 \times 3 \times 3$ | $48@1 \times 3 \times 3$ | $64@1 \times 2 \times 2$ | $64@1 \times 2 \times 2$ | $72@1 \times 2 \times 1$ |

Figure 8a illustrates that the amount of value information implied by the input data increased with the day number $D$ covered by the HLM, so that the model could learn more favorable features to estimate future loads. However, after the covered days reached a certain number, continuing to increase $D$ did not bring significant benefits to the accuracy, such as $28 \times 12$ vs. $21 \times 12$. The reason was that the historical load data far from the forecasted hour had a weakened temporal correlation with the forecast values, and thus, the positive contribution to the inferred task was less or even negative. In addition, more days meant more complex calculations. While a smaller matrix size slightly reduced the forecasting performance, it was beneficial to the implementation efficiency of our scheme. For the tradeoff, the load matrix size was set to $14 \times 12$ in our cases.

Second, we tested the influence of the frame number on the RECO model. In Figure 8b, fewer frames led to poor forecasting accuracy and stability. Similar to the previous analysis, for the learned feature sequences from the frames that were farther from the forecasted hour. the memory information generated in the GRU had a less positive impact on the regression results. Therefore, simply increasing the frame number usually did not lead to performance improvement.

Finally, Table 2 lists the quantified performance of different model structures. It can be seen that the learning module that joined the pooling operation, especially the maximum pooling, deteriorated the accuracy and stability. Unlike the classification problems, each element of the formulated HLM was

usually informative, and thus, the spontaneous convolution exploration was more desirable for learning suitable features. Additionally, a smaller kernel was not conducive to improving the model performance due to the data redundancy of the created HLT. For the regression module, more nodes or more hidden layers may incur over-fitting and gradient problems in training due to large load variances.

**Table 2.** Quantified performance of different model structures.

| MAE/RMSE　　GRU<br>CNN | 1 Layer (128) | 1 Layer (256) | 1 Layer (512) | 2 Layer (128) | 3 Layer (128) |
|---|---|---|---|---|---|
| Figure 3, mean pooling | 4.32/6.26 | - | - | - | - |
| Figure 3, max pooling | 5.51/7.72 | 5.00/7.93 | - | - | - |
| $1 \times 2 \times 2$, no pooling | 3.88/5.05 | 3.82/5.10 | 3.88/5.17 | - | - |
| Figure 3, no pooling | 3.84/4.80 | 3.79/4.96 | 3.91/5.20 | 3.82/5.12 | 4.12/6.07 |

The data are from the UGI in 2018, and the unit of MAE or RMSE is MW. "-" stands for untested. The convolution stride for three directions with the pooling operation is set to [1, 1, 1] and a one-half down-sampling.

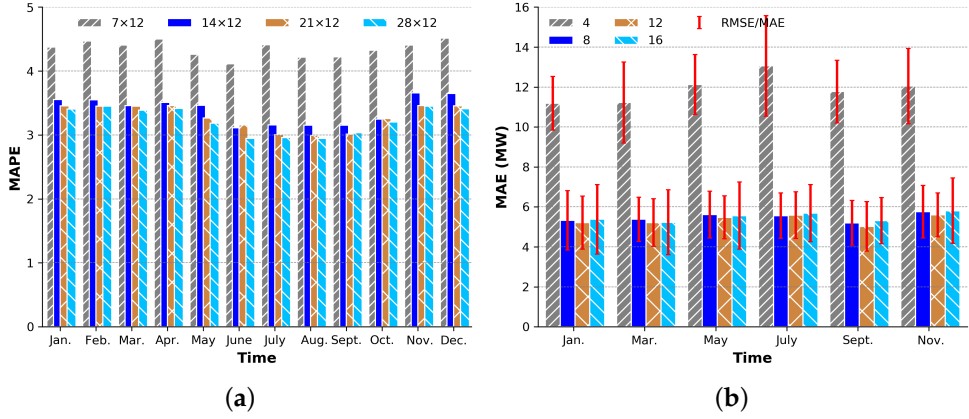

(**a**)　　　　　　　　　　　　　　　　　(**b**)

**Figure 8.** (**a**) Forecast errors of historical load matrices with different sizes. The data are from the PAPER in 2018, and the frame number is eight. (**b**) Influence of different time frames on the forecast results. The data are from the RECO in 2017, and "4, 8, 12, 16" indicate the frame number of the HLT.

*3.4. Performance Comparison*

3.4.1. Data Preprocessing Algorithm

The DALM algorithm based on load spatiotemporal correlation was an important part of our forecasting scheme, and thus, we first evaluated the benefits of the low rank decomposition of the raw load tensor to the model. The forecast curves brought by different processing combinations are shown in Figure 9.

Taking the best results obtained by our plan as the baseline, the input data composed of the raw load plus the gradient matrix (or HLM + FGM + DGM) yielded the worst estimate; without the load change information (or BLM + FLM), the model performance did not significantly decline, which was attributed to the fact that the created HLM contained sufficient information and thus compensated the lack of gradient information. These results showed that the proposed DALM could effectively untangle the cyclical components and fluctuations in the load data, which made the load pattern represented by the input data more hierarchical, thereby helping the 3D CNN to extract more targeted features.

Further, we compared the DALM with two other commonly used load decomposition algorithms, namely WT [6] and VMD [8]. We used the fast discrete WT based on the Mallat algorithm to decompose each column of the HLM $L^t$ into one approximation component and three detail components independently. Then, the approximation components of each column were reorganized into one approximation matrix in the original order, and three detail components performed the same operation to form three detail matrices. Finally, the above four formed an input tensor at hour $t$. The VMD performed transformation similar to the above process on the HLM, where we set five intrinsic mode functions and one residue to form an input tensor.

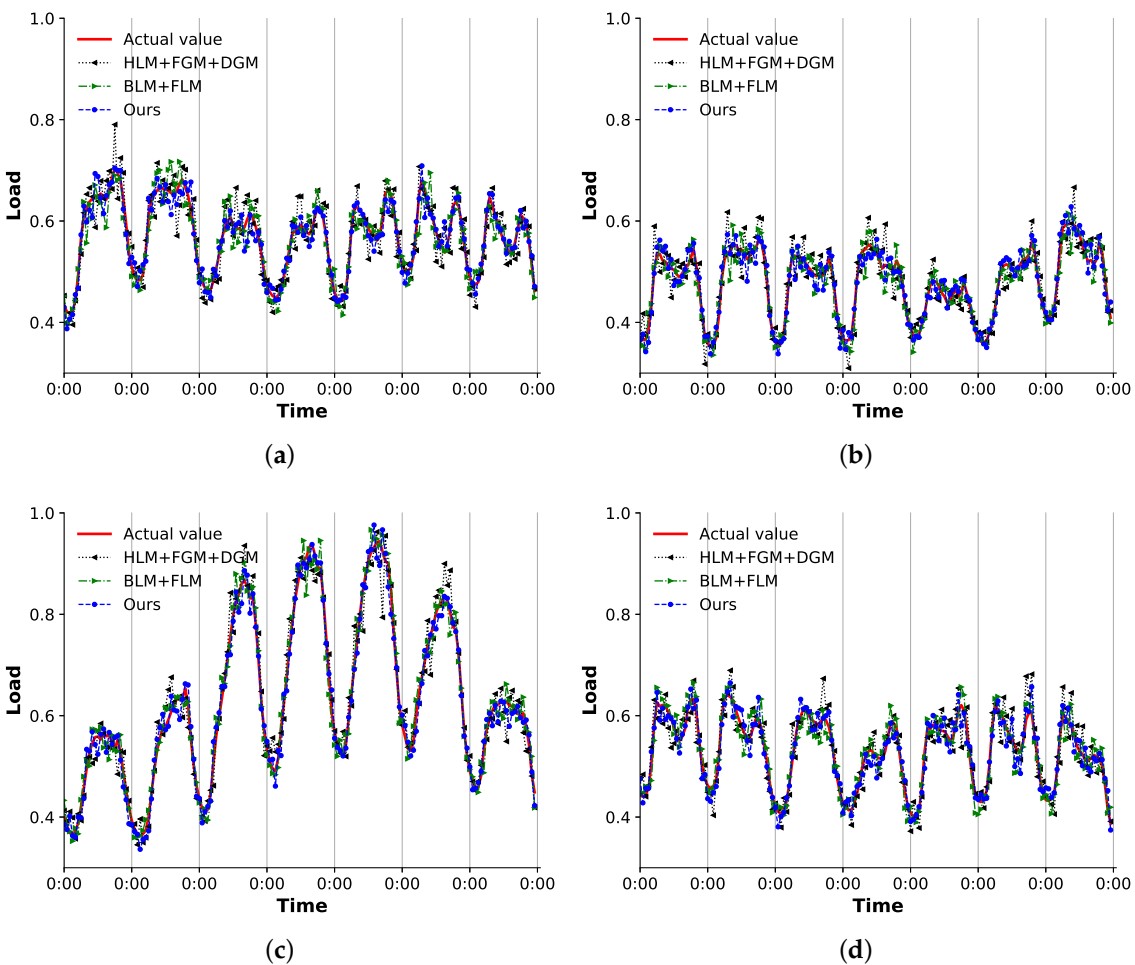

**Figure 9.** Forecast results for different input data patterns. The data are from the UGI in (**a**) February, (**b**) April, (**c**) August, and (**d**) October in 2018, and the results are not denormalized.

In Figure 10, under the same forecasting model, our DALM achieved better results than the two one-dimensional deterministic decomposition algorithms. The reasons were mainly twofold: (1) For the load data with strong uncertainty, although the WT and VMD algorithms had strong physical meanings, the decomposition mechanism based on the deterministic fundamental components made them, especially the WT, unable to adapt to the randomness and volatility of the data well. (2) The proposed low rank decomposition algorithm relied on a large number of iterations to realize a comprehensive consideration

of the relevant information between matrix elements, which not only effectively separated structured data and noise, but could counteract the data fluctuation by adapting to the differences of the objectives.

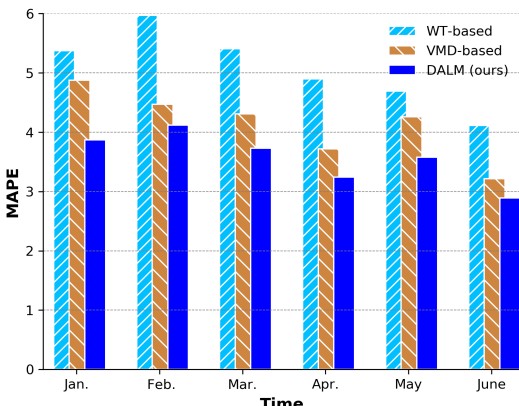

**Figure 10.** Forecast errors with different decomposition algorithms. The data from the UGI in 2019.

### 3.4.2. Comparison of the Overall Scheme

To show the overall superiority of our proposed scheme, we compared it with several advanced techniques of its kind. The results are shown in Figure 11 and Tables 3–5.

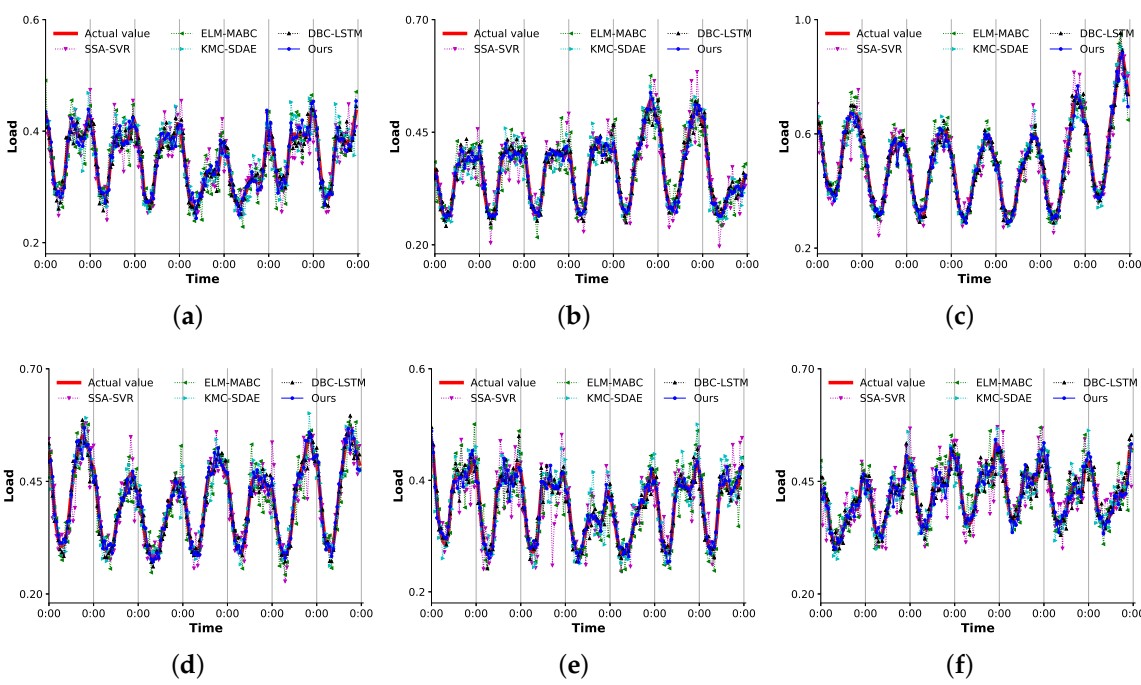

**Figure 11.** Forecast results for different models and the data from the RECO in (**a**) February, (**b**) April, (**c**) June, (**d**) August, (**e**) October, and (**f**) December 2017.

In general, our proposal demonstrated better accuracy and stability in Figure 11, while the forecast results of SSA-SVR and ELM-MABC had large errors and variances. For the more volatile cases, as shown in Figure 11a,f, ours and DBC-LSTM still guaranteed a forecast curve that roughly matched the actual load. In the relatively stable scenarios (see Figure 11c,d), the forecast error of the KMC-SDAE was

significantly reduced; the SSA-SVR and ELM-MABC, however, failed to have satisfactory inference, especially the SSA-SVR.

Further, with the MAE and the RMSE, we exhibited the quantified performance of each mode and analyzed the reasons for these results. According to the two average criteria listed in Table 3, the proposed forecasting scheme exceeded these previous methods in terms of overall performance.

Specifically, the SSA-SVR modeled the hourly load independently and carefully classified the raw data, including weather and holidays, with a complex feature selection algorithm. However, since the load change and input variables were not absolute causal links, the pseudo-nonlinear mapping of the employed radial basis function caused its $\varepsilon$-SVR to lack adaptability for strong fluctuation scenarios. This major drawback resulted in poor performance. For the ELM-MABC, although one-dimensional WT and swarm intelligence algorithm based weight optimization were applied to build input data and the ELM model, the linear separation approach and single hidden layer networks with 20 nodes were not sufficient to characterize the load estimation with a strong random nature. For the KMC-SDAE, we used real temperature information instead of estimated data. However, the layer-by-layer feature extraction mechanism of the SDAE for one-dimensional input did not cope well with complex electricity consumption scenarios. Of these models, the DBC-LSTM demonstrated performance comparable to ours, especially against the background of a smooth load, which was mainly due to its full use of the natural advantages of LSTM for time series forecasting. In addition, it applied a density based clustering algorithm to analyze and classify load cases with strong volatility and then made the input data patterns with difference, which benefited the forecasting performance as well.

**Table 3.** Forecast errors of different models. SDAE, stacked denoising autoencoder.

| Model / Time | SSA-SVR | | ELM-MABC | | KMC-SDAE | | DBC-LSTM | | Ours | |
|---|---|---|---|---|---|---|---|---|---|---|
| | MAE | RMSE | MAE | RMSE | MAE | RMSE | MAE | RMSE | MAE | RMSE |
| January | 7.33 | 16.22 | 6.44 | 11.16 | 5.76 | 10.67 | 3.04 | 5.47 | 2.10 | 3.05 |
| February | 10.42 | 24.32 | 7.52 | 13.52 | 4.74 | 8.99 | 2.33 | 3.99 | 2.26 | 2.79 |
| March | 8.09 | 17.86 | 5.00 | 9.04 | 5.39 | 10.10 | 2.71 | 4.33 | 2.21 | 3.09 |
| April | 7.21 | 15.36 | 5.35 | 9.47 | 6.82 | 11.50 | 1.95 | 3.26 | 2.01 | 2.81 |
| May | 6.21 | 13.59 | 5.98 | 9.12 | 4.02 | 7.04 | 2.62 | 4.04 | 1.90 | 2.62 |
| June | 8.91 | 16.79 | 6.03 | 10.27 | 3.78 | 6.61 | 2.86 | 3.29 | 2.15 | 2.52 |
| July | 8.10 | 17.58 | 5.33 | 8.46 | 3.60 | 5.53 | 1.99 | 3.03 | 2.02 | 2.23 |
| August | 7.22 | 12.29 | 4.01 | 6.61 | 4.44 | 6.71 | 1.84 | 2.72 | 1.85 | 2.11 |
| September | 7.01 | 15.51 | 5.42 | 9.18 | 4.66 | 7.77 | 2.66 | 4.31 | 1.95 | 2.49 |
| October | 8.02 | 17.81 | 6.07 | 10.86 | 7.03 | 10.61 | 2.44 | 3.46 | 2.55 | 3.61 |
| November | 8.00 | 18.39 | 6.92 | 13.06 | 6.76 | 12.33 | 2.59 | 4.64 | 2.34 | 2.84 |
| December | 9.09 | 20.56 | 7.10 | 13.11 | 6.03 | 10.98 | 2.93 | 4.79 | 2.38 | 2.89 |
| **Mean** | 7.97 | 17.19 | 5.93 | 10.32 | 5.25 | 9.07 | 2.51 | 3.95 | 2.14 | 2.76 |

The data are from the PAPER, and all are normalized data; the unit of the MAE and RMSE is "%".

Table 4 lists the quantified errors of each model when performing load inference for the absolute forecasting period. It can be seen that ours still highlighted the superiority over other models in terms of generalization and achieved a relative minimum error of 2.97. As mentioned above, the reason lied in our multidimensional input data planning and the spatiotemporal feature learning based variable acquisition for the regression module. Obviously, there were certain gaps between the selected methods and our forecasting scheme.

Finally, we compared our proposal with two methods that considered the spatiotemporal correlation of loads, and Table 5 lists the overall quantified performance of the three. Referring to Tables 3 and 4, while the traditional deterministic models were used, the sufficient mining of the correlation helped the

NCST-LF and knmV-AR to have good stability during the absolute forecasting period. This suggested that the spatiotemporal correlation of loads was of great value for improving the forecasting performance of the model. However, the deterministic models suffered from weak descriptions of load randomness and volatility, which made them struggle to improve overall performance significantly, especially in the case of large load fluctuations.

**Table 4.** Forecast errors of different models.

| MAPE \ Model Time | SSA-SVR | ELM-MABC | KMC-SDAE | DBC-LSTM | Ours |
|---|---|---|---|---|---|
| January | 10.10 | 9.13 | 10.17 | 4.26 | 3.95 |
| February | 12.09 | 11.09 | 8.09 | 4.53 | 4.05 |
| March | 11.11 | 9.11 | 10.11 | 4.10 | 3.85 |
| April | 9.90 | 10.9 | 9.90 | 3.91 | 3.62 |
| May | 9.80 | 8.80 | 7.80 | 4.00 | 3.11 |
| June | 10.70 | 7.24 | 8.66 | 3.93 | 2.97 |
| **Mean** | 10.61 | 9.38 | 9.12 | 4.12 | 3.59 |

The data are from the PAPER in 2019.

**Table 5.** Forecast errors of different spatiotemporal correlation based methods

| M/R \ Model Time | January | February | March | April | May | June | Mean |
|---|---|---|---|---|---|---|---|
| **NCST-LF** | 7.76/12.65 | 8.14/13.10 | 7.99/12.96 | 6.73/10.50 | 7.04/11.22 | 6.27/8.29 | 7.32/11.45 |
| **knmV-AR** | 7.06/10.15 | 6.60/9.44 | 6.72/9.98 | 7.17/10.72 | 6.26/9.16 | 5.93/8.46 | 6.63/9.65 |
| **Ours** | 2.66/3.41 | 2.73/3.43 | 2.65/3.50 | 2.30/2.92 | 2.19/2.77 | 2.03/2.53 | 2.42/3.11 |

The data are from the UGI in 2019, and all are normalized data. "M/R" denotes MAE/RMSE, and the unit of the MAE and RMSE is "%".

## 4. Conclusions

In the context of the EI, we focused on one-hour-ahead load forecasting and attempted to improve the forecasting performance by leveraging a novel data driven scheme. First, the spatiotemporal correlation based multidimensional input automatically incorporated a variety of information related to load changes, which was equivalent to providing the forecasting model with a nearly redundant feature base. Second, data preprocessing based on low rank decomposition and load gradients made the features refined and hierarchical, which was conducive to learning more targeted features. Third, the developed 3D CNN-GRU model could spontaneously capture the desired features with time attributes and then map them to future loads through nonlinear regression with memory. Finally, our scheme achieved good accuracy (mean absolute error of 2.14%) and stability (root mean squared error of 2.76%). Given such a performance, the proposed method was suitable for consumers with large fluctuations in load curves and large differences in aggregation scales, such as power distribution optimization cases in the industrial fields such as steel, metallurgy, and machinery manufacturing.

Extensive experimental results showed the effectiveness of the proposed scheme and demonstrated the value and significance of the spatiotemporal correlation of distributed energy units for energy interconnection research. Nevertheless, there were still some open problems in this study. For instance, this study did not attempt to add more types of valuable factors, such as temperature and humidity, to the

input data for more reliable results; also, more advanced feature learning paradigms need to be explored to discover data patterns more precisely. These will be the focus of our further study.

**Author Contributions:** L.D. built the forecasting model and completed the manuscript; L.Z. designed the optimization algorithm and revised the manuscript; X.W. collected the data and performed the main experiments. All authors read and agreed to the published version of the manuscript.

**Funding:** This work was supported in part by the National Natural Science Foundation of China under Grant 61771258, in part by the Postgraduate Research and Practice Innovation Program of Jiangsu Province under Grant KYCX18_0884, and in part by the Anhui Science and Technology Department Foundation under Grant 1908085MF207.

**Conflicts of Interest:** The authors declare no conflict of interest.

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
