# Peer review of "Spatiotemporal Feature Learning Based Hour-Ahead Load Forecasting for Energy Internet"

_electronics, doi:10.3390/electronics9010196_

Round 1
Reviewer 1 Report
the paper presents a load forecasting method as a tool for the management of energy internet. To this end, a deep neural network based approach is used. The training is performed on the only data records of power, rather than using other related variables. The spatiotemporal correlation of series makes it possible to strongly reduce the volume of data, with the benefit of the computational burden. The suitability of the proposed approach is evaluated on a well known benchmark, and the performance is compared with that of other approaches retrieved from the literature
Remarks
The only doubt of the approach is related to the short-term prediction performed by the training system. It would be necessary to indicate which specific needs in the management of the electricity grid are met with such a short time horizon.
On the other hand, it is reasonable to assume that aggregating a relevant number of customers, the scatter of the series is low and a good prediction in a short term could be obtained also by a linear filter.
Finally, some indications should be given on the degree of precision of the prediction that would be suitable for the scope of the application
#Minor points:
>line 149: double verb
>Figures 5, 9, and 10: the time axis is constantly zero
>line 272: article "an" for "a"
Author Response
Dear Reviewers and Editors:
We are very grateful for the positive and constructive comments from reviewers and editors on our manuscripts, which are very helpful for improving the academic value of our work.
We have studied the reviewers’ feedback carefully and have tried our best to improve our manuscript according to the comments.
The following are the responses to the comments and some explanations for the revised manuscript.
Please note that all revisions to comments were marked blue in the revised manuscript.
Kind Regards,
The authors
Point 1: The only doubt of the approach is related to the short-term prediction performed by the training system. It would be necessary to indicate which specific needs in the management of the electricity grid are met with such a short time horizon.
Response 1: The forecasting scheme we propose belongs to the category of ultra short-term load forecasting, which infers the load trend of the next hour. Ultra short-term forecasting is usually used for power quality control, online operation safety monitoring and prevention, and emergency control.
Since the proposed forecasting model (i.e., 3D CNN-GRU) performs offline training, and the trained model does not need to be retrained by adding new samples in the short term.
Please refer to Tab. 4 in the Subsection 3.4.2, and our trained models using 2012-2018 sample still have a good inference on the load trend in Jan.-June 2019.
We have supplemented explanations in the revised manuscript.
Please see lines 92-96.
Point 2: On the other hand, it is reasonable to assume that aggregating a relevant number of customers, the scatter of the series is low and a good prediction in a short term could be obtained also by a linear filter.
Response 2: In this work, we selected 12 geographically close zones from the PJM to represent distributed aggregation customers.
The load aggregation levels of these assumed customers are different. For example, the largest PS zone has a range [3200,8300], the median DUQ is [1027,2470], and the smallest UGI is [67,210] (2019,1-6 statistics). The three zones we used to evaluate the test results: PAPWR, RECO, and UGI, whose load ranges (in MW) are [204.4, 1,053.8], [83.1, 438.4], and [64.7, 223.5] respectively.
Although the load aggregation levels in different assumed customers are inconsistent,
we focus on inferring the load curve trend and all raw values need be quantified,
it thus has small impact on the theoretical evaluation of the proposed forecasting scheme.
In addition, load aggregation at different scales is more in line with the characteristics of distributed units, and can better reflect the generalization of the proposed model, which will be demonstrated in experimental results.
We have supplemented the description about this problem in the revised manuscript.
Please see lines 144-152.
Point 3: Finally, some indications should be given on the degree of precision of the prediction that would be suitable for the scope of the application.
Response 2: Our scheme achieves good accuracy (mean absolute error of 2.14%) and stability (root mean square error of 2.76%). Given such a performance, the proposed method is suitable for consumers with large fluctuations in load curves and large differences in aggregation scales, such as power distribution optimization cases in the industrial fields such as steel, metallurgy, and machinery manufacturing.
We have supplemented explanations in the revised manuscript.
Please see the Conclusions section (lines 394-397).
Point 4: line 149: double verb.
Response 2: We have revised this error in the revised manuscript.
Please see line 179.
Point 5: Figures 5, 9, and 10: the time axis is constantly zero
Response 2: The “0:00” indicates the beginning of a natural day.
In these three figures, two adjacent vertical dashed lines represent the load curve of a natural day, that is, from 0:00 to 23:00.
However, due to the size of the simulation graph, we only show the starting hour of a day.
We have supplemented this description in the Fig. 5.
Point 6: line 272: article "an" for "a"
Response 2: We have revised this error in the revised manuscript.
Please see line 306.
In addition, we have edited the entire manuscript with the help of a professional English editing agency. The relevant English editing certificate (as below) has been uploaded to the system along with the revised manuscript.

Reviewer 2 Report
The authors develop a methodology to predict short-term electricity load by using a spatio-temporal forecasting model on univariate electricity load data from the US. They construct a three dimensional input tensor, that is constructed by the dimensions hour, day, and region. They thereby integrate spatial correlations as information to their forecasting model. They then feed this tensor into a 3D CNN-GRU prediction model. The results show to be promising. However, some major and minor concerns have to be addressed:
I have doubts regarding your assumption in lines 82-84, that “all external factors are ultimately reflected in the load profile”. I agree that there are good arguments for univariate prediction, however, external factors such as environmental influences are valuable inputs for load prediction. Please discuss your assumption and present further arguments. Could you please clarify why you choose to construct the HLT as a three-dimensional tensor with two temporal dimensions (day, hour + zone) instead of using a two-dimensional Tensor (hour + zone) I suggest that you restructure your comparison (validation) of your model to other SOTA models in a more systematic approach. The contribution of your approach would become clearer if you compare your model (1) against models, that do not use spatial correlations and (2) against models that do already use spatial correlations (see your reference #31 or the following reference: J. Xu, Meng Yue, D. Katramatos and S. Yoo, "Spatial-temporal load forecasting using AMI data," 2016 IEEE International Conference on Smart Grid Communications (SmartGridComm), Sydney, NSW, 2016, pp. 612-618. doi: 10.1109/SmartGridComm.2016.7778829). By doing so, you could show (1) the advantages of spatiotemporal approaches in general and (2) the advantages of your model against similar models in particular. I do not quite see the connection between the term Energy Internet and your particular forecasting model (why is your model tailored to Energy Internet?). Could you please elaborate on this and justify, why the term Energy Internet is so dominant in your title.Additionally, there are some minor points comprising the following:
Please avoid using abbreviations without introducing them in the abstract (see SOTA) Please introduce your understanding of the term “drive data”. In addition please specify what aggregation level your load curves have and what spatial area a “zone” covers Please revise Figure 1, e.g. the decision blocks of your flow chart should have two conditions Following lines 87-88, please explicitly name the three input dimensions that you refer to Introduce the abbreviation PJM and add further information on the data set in your text In the caption of Figure 4 you describe a typical size of the HLT as 8x14x12. As I understand it, the HLT consists of the dimensions day, hour, and zone. Besides the 12 zones that you refer to I do not see the origin of the other two values In lines 198-199, please clarify what you are referring to by “the number of the HLT is the same as the label” Some language flaws are present, please revise the manuscript, ideally with the help of a native tongue speakerThe presented forecasting model could be a valuable contribution to the research field. Overall, the content is well presented. However, some concerns must be addressed before I can recommend the paper for publication.
Author Response
Dear Reviewers and Editors:
We are very grateful for the positive and constructive comments from reviewers and editors on our manuscripts, which are very helpful for improving the academic value of our work.
We have studied the reviewers’ feedback carefully and have tried our best to improve our manuscript according to the comments.
The following are the responses to the comments and some explanations for the revised manuscript.
Please note that all revisions to comments were marked blue in the revised manuscript.
Kind Regards,
The authors
Point 1: I have doubts regarding your assumption in lines 82-84, that “all external factors are ultimately reflected in the load profile”. I agree that there are good arguments for univariate prediction, however, external factors such as environmental influences are valuable inputs for load prediction. Please discuss your assumption and present further arguments.
Response 1: A daily load curve typically consists of (1) the cyclical load (accounting for a relatively large proportion) in the regular pattern, (2) the uncertain load (small proportion) caused by external factors such as weather, holidays, and and customer behaviour, and (3) the noise that cannot be physically explained (minimum proportion), all of which are ultimately quantified and superimposed as a load sequence. Sufficient load sequences represent a variety of data patterns, and thus reliable forecasts can be achieved by mining large amounts of highly correlated load data only, which has also been confirmed in several recent studies (References [8,19,21]).
Due to the weak representation ability or nonlinear mapping ability, many models have to use more types of input variables to make up for the neglect of the fluctuating data pattern.
In this work, we designed the input data scheme combined with the 3D CNN-GRU model with good representation learning and regression capabilities, which can effectively avoid these problems (please refer to the results of the evaluation).
In addition, just like the reviewers' comments, in practical applications, external factors such as temperature, humidity, and wind are valuable inputs for load forecasting. If these factors are considered, the proposed method can provide better forecasting results. However, given the length and workload of a manuscript, we hope to reconsider these problems in our further study (mentioned in the conclusions).
Our description about this problem in the initial manuscript is indeed unclear, so we have conducted a detailed analysis and explanation in the revised manuscript.
Please see lines 78-91, 400-403.
Point 2: Could you please clarify why you choose to construct the HLT as a three-dimensional tensor with two temporal dimensions (day, hour + zone) instead of using a two-dimensional Tensor (hour + zone) .
Response 2: To make full use of the spatiotemporal correlation of historical load data, the temporal correlation here refers to the load data between consecutive days, we construct a three-dimensional load tensor( HLT: N × D × Z) with three dimensions corresponding to hour dimension (N), day dimension (D), and zone dimension (Z). The tensor elements are historical load values.
Specifically,the zone dimension is fixed and denotes the 12 selected geographically close zones;the day dimension denotes the number of consecutive days at a fixed hour;the hour dimension denotes the number of consecutive hours before the forecasted hour.
There are two main reasons for constructing the input data in this way:
(1) Due to similar external factors (such as weather) between adjacent daily loads, there is a high temporal correlation between consecutive daily load trends. We analyzed and explained this in 2.1.2. Considering more daily load can provide more value features for the regression of forecasting model.
(2) 3D CNN has the ability to independently explore three input channels (2D CNN performs convolution on multi-channel input and then adds), and can well learn the temporal correlation between consecutive daily load data.
Since our presentation was simple in the initial manuscript, we have supplemented explanations in the revised.
Please see lines 139-141, 155-156.
Point 3: I suggest that you restructure your comparison (validation) of your model to other SOTA models in a more systematic approach. The contribution of your approach would become clearer if you compare your model (1) against models, that do not use spatial correlations and (2) against models that do already use spatial correlations (see your reference #31 or the following reference)
Xu, Meng Yue, D. "Spatial-temporal load forecasting using AMI data," 2016 IEEE International Conference on Smart Grid Communications, pp. 612-618. doi: 10.1109/SmartGridComm.2016.7778829.By doing so, you could show (1) the advantages of spatiotemporal approaches in general and (2) the advantages of your model against similar models in particular.
Response 3: Regarding recommendation (1), we have conducted comparisons with similar advanced models in the “3.4. Performance comparison”, and these models do not involve the spatiotemporal correlation in load changes, so we think these comparisons can be regarded as "against models that do not use spatial correlations".
Regarding recommendation (2), we conducted the comparison from two aspects:
1) Our overall scheme was compared with the recommended method ([32]) and reference [33], both of which consider the spatiotemporal correlation of the load. The two methods used the same original load data as ours;
2) Based on the 3D CNN-GRU model, we evaluate the prediction errors of the proposed (spatialtemporal correlation-based) low-rank decomposition algorithm and other preprocessing algorithms.
Reference [32] only uses K-clustering and PCC to analyze and cluster load units with spatiotemporal correlation to facilitate the prediction model, and does not involve specific data preprocessing algorithms. Similarly, reference [33] utilizes the spatiotemporal correlation of the load to perform prediction based on Compressive Sensing and the multivariate autoregressive model, and does not involve preprocessing algorithms as well.
Therefore, we chose two other common preprocessing algorithms, wavelet transform and variational mode decomposition, aiming to highlight the advantages of our preprocessing algorithm.
The test data are from the first six months of 2019 of UGI, and we quantified the overall performance of the three models using the MAE , RMSE, and MAPE.
Meanwhile, we adjusted the content structure of 3.3 and 3.4 accordingly.
The final test results have been added to the revised manuscript.
Please see subsection 3.4.1, 3.4.2 (in blue), Fig. 10 and Tab. 5. ,lines 246-250.
Point 4: I do not quite see the connection between the term Energy Internet and your particular forecasting model (why is your model tailored to Energy Internet?). Could you please elaborate on this and justify, why the term Energy Internet is so dominant in your title.
Response 4: Energy Internet is a new type of information-energy integration power system architecture based on the concept of the Internet. Under this architecture, there will be a large number of distributed microgrids. Our objects are those distributed microgrids, whose load changes have a spatiotemporal correlation because of geographic proximity.
While the Energy Internet is still in the theoretical and small-scale pilot phase, its concept has appeared for more than a decade. So, we have not introduced it too much, and the related detailed introduction can refer to the listed literatures [1-4].
Since our statement of about this was not accurate enough, we have modified it in the revised manuscript.
Please see lines 25-27.
Point 5: Please avoid using abbreviations without introducing them in the abstract (see SOTA) 。
Response 5: We have revised this in the abstract.
Point 6: Please introduce your understanding of the term “drive data”. In addition please specify what aggregation level your load curves have and what spatial area a “zone” covers.
Response 6: The term "drive data" in our manuscript refers to the model "input data", for corresponding to the term "data-driven approach".
In order to regulate the expression, we have changed this phrase in the full text to "input data".
Because the revisions are too scattered, we didn't mark them in blue.
In our case studies, we used the real-world Metered Load Data from the PJM, and the 12 zones selected were geographically close together, which can be queried from two reference links ([34] and [35]) respectively.
The load aggregation levels of these 12 zones are different. For example, the largest PS zone has a range [3200,8300], the median DUQ is [1027,2470], and the smallest UGI is [67,210] (2019,1-6 statistics).
The three zones we used to evaluate the test results: PAPWR, RECO, and UGI, whose load ranges (in MW) are [204.4, 1,053.8], [83.1, 438.4], and [64.7, 223.5] respectively.
Although the load aggregation levels in different zones are inconsistent, we focus on inferring the trend of the load curve, and all original values need be quantified based on their maximum load, so it has small impact on the theoretical evaluation of the proposed method.
In addition, load aggregation at different levels is more in line with the characteristics of distributed units, and it can better reflect the generalization of the proposed, which is shown in the experimental results.
We have supplemented the description about this problem in the revised manuscript.
Please see lines 144-152, 225-226, 232.
Point 7: Please revise Figure 1, e.g. the decision blocks of your flow chart should have two conditions.
Response 7: Based on the overall structure of our work, we have adjusted the flow chart of Fig. 1 in the revised manuscript.
Point 8: Following lines 87-88, please explicitly name the three input dimensions that you refer to.
Response 8: These three input dimensions are named depth, width, and height in TensorFlow. We have added it in the revised manuscript.
Please see line 107.
Point 9: Introduce the abbreviation PJM and add further information on the data set in your text.
Response 9: PJM is a regional transmission organization in the U.S. that currently operates an electric transmission system serving all or parts of the eastern 13 states and the Washington, D.C., covering 29 electricity zones including the above 12.
We have supplemented the description of the “PJM” and the selected zones in the revised manuscript.
Please see lines 144-152.
Point 10: In the caption of Figure 4 you describe a typical size of the HLT as 8x14x12. As I understand it, the HLT consists of the dimensions day, hour, and zone. Besides the 12 zones that you refer to I do not see the origin of the other two values.
Response 10: The first two dimensions of HLT are hour and day, which can be set arbitrarily in theory according to the predicted needs (please see Tab. 1 and Fig. 8). The 8x14x12 we give is specifically set according to the 3D CNN structure of Fig. 4.
Since the 3D CNN network parameters corresponding to different HLT (or HLM) sizes are not exactly the same (please see Tab. 1 and Fig. 8), while the 3D CNN structure in Fig. 4 is specific, e.g., the first 3D filter-bank is "32 (group) @ 1 (depth) × 3 (height) × 3(width)", and the most suitable HLT (or HLM) size for this structure is 8x14x12 in our case study.
Since our interpretation of this problem was not clear enough, we have supplemented it in the revised manuscript.
Please see lines 139-141, 155-156, and the caption of Fig. 4.
Point 11: In lines 198-199, please clarify what you are referring to by “the number of the HLT is the same as the label”.
Response 11: Since our model training performs a supervised learning, an input data for training needs to correspond to an actual load label.
Similarly, to facilitate testing the trained model, we also set up a label for each test data.
Therefore, the labels number and the input number are the same.
Indeed, our representations in the initial manuscript are inaccurate and ambiguous, and we have modified and supplemented the representations.
Please see lines 190-193,216, 229.
Point 12: Some language flaws are present, please revise the manuscript, ideally with the help of a native tongue speaker.
Response 12: We have edited the entire manuscript with the help of a professional English editing agency. The relevant English editing certificate has been uploaded to the system along with the revised manuscript.
Since there are many changes, we didn't mark them in blue.

Round 2
Reviewer 2 Report
The authors have addressed the comments thoroughly. There are some minor concerns left before I can fully recommend the work for publication:
Please revise the initial sentences in your abstract. First, you claim that there are challenges and favorable circumstances of EI for load forecasting but then miss naming them explicitly. This will help the reader to understand you motivation. In Fig. 10, please label your algorithm with its name, i.e. DALM (e.g. DALM (ours))Author Response
Dear Reviewers and Editors:
We are very grateful to the reviewers for reviewing our manuscript again, and these comments are very helpful to further improve our work. We have completed the revision based on the comments.
Here are some explanations of the revised draft.
Kind Regards,
The authors
Point 1: Please revise the initial sentences in your abstract. First, you claim that there are challenges and favorable circumstances of EI for load forecasting but then miss naming them explicitly. This will help the reader to understand you motivation.
Response 1: According to the reviewer's recommendations, and considering that the the initial sentence and the entire abstract are not very relevant, we have deleted it in the Abstract of the revised manuscript.
Please see the Abstract of the revised manuscript.
Point 2: In Fig. 10, please label your algorithm with its name, i.e. DALM (e.g. DALM (ours))
Response 2: According to the reviewer's suggestion, we have revised the label of the DALM algorithm in Fig. 10.
Please see Fig. 10.
